# Hospital Chaplain Burnout, Depression, and Well-Being during the COVID-19 Pandemic

**DOI:** 10.3390/ijerph21070944

**Published:** 2024-07-19

**Authors:** Patricia K. Palmer, Zainab Siddiqui, Miranda A. Moore, George H. Grant, Charles L. Raison, Jennifer S. Mascaro

**Affiliations:** 1Spiritual Health, Woodruff Health Sciences Center, Emory University, Atlanta, GA 30322, USA; pkpalme@emory.edu (P.K.P.); ghgrant@emory.edu (G.H.G.); raison@wisc.edu (C.L.R.); 2Department of Family and Preventive Medicine, Emory University School of Medicine, Atlanta, GA 30322, USAmiranda.moore@emory.edu (M.A.M.); 3Department of Medicine, Emory University School of Medicine, Atlanta, GA 30322, USA

**Keywords:** chaplain, burnout, spiritual health, COVID-19, well-being, depression

## Abstract

Healthcare personnel experienced unprecedented stressors and risk factors for burnout, anxiety, and depression during the COVID-19 pandemic. This may have been particularly true for spiritual health clinicians (SHCs), also referred to as healthcare chaplains. We administered a daily pulse survey that allowed SHCs to self-report burnout, depression, and well-being, administered every weekday for the first year of the pandemic. We used a series of linear regression models to evaluate whether burnout, depression, and well-being were associated with local COVID-19 rates in the chaplains’ hospital system (COVID-19 admissions, hospital deaths from COVID-19, and COVID-19 ICU census). We also compared SHC weekly rates with national averages acquired by the U.S. Census Bureau’s Household Pulse Survey (HPS) data during the same timeframe. Of the 840 daily entries from 32 SHCs, 90.0% indicated no symptoms of burnout and 97.1% were below the cutoff for depression. There was no statistically significant relationship between any of the COVID-19 predictors and burnout, depression, or well-being. Mean national PHQ-2 scores were consistently higher than our sample’s biweekly means. Understanding why SHCs were largely protected against burnout and depression may help in addressing the epidemic of burnout among healthcare providers and for preparedness for future healthcare crises.

## 1. Introduction

Healthcare personnel experienced unprecedented stressors and risk factors for burnout during the COVID-19 pandemic. Although the prevalence of burnout varied widely depending on heterogenous contextual factors that confer protection from, or risk of, burnout [1], physicians and nurses working in intensive care reported higher burnout rates than rates observed prior to the pandemic [2]. Spiritual health clinicians (SHCs), also referred to as healthcare chaplains, are employed in approximately two-thirds of all hospitals in the United States [3,4] and work collaboratively with other healthcare professionals to recognize and respond to the emotional, psychosocial, spiritual, and moral distress of patients, their loved ones, and healthcare providers and staff [3,4,5,6,7]. While some hospital systems prohibited SHCs from working on COVID-19 units or within the hospital as a whole during the pandemic, other SHCs provided important care to vulnerable and high-acuity patients, their families, and the staff who cared for them [8]. However, little research has examined the mental and professional well-being of SHCs providing vital frontline spiritual health care during the pandemic.

Although previous research indicates that SHCs may be particularly resilient in their ability to maintain low levels of burnout and high levels of compassion satisfaction even while providing care to high-acuity patients experiencing profound distress [9,10,11,12,13], the pandemic had the potential to be particularly harmful to SHCs for several reasons. First, protocol and healthcare system changes removed or reduced many of the interpersonal factors that are protective to hospital chaplains. For example, SHCs reported that safety protocols to mitigate risk during the pandemic often increased disconnection and depersonalization [14]. In addition, SHCs provide an often underappreciated level of care and support for other clinical staff in distress, which increased during the pandemic [14]. A third risk factor experienced by SHCs was high levels of professional uncertainty [8], and many chaplains reported feeling unsure of their role in the context of the crisis [15].

Characterizing changes in burnout and well-being among SHCs involved in direct patient care during the pandemic may provide novel perspectives on how to optimize the personal and professional wellness of all members of the interprofessional healthcare team. With this goal, we attempted to answer three questions. First, how did SHCs rate their level of burnout, depression, and well-being during the first year of the pandemic? Second, were rates of burnout, depression, and well-being associated with local COVID-19 rates in the SHC’s hospital system? And third, to contextualize SHC mental health during the pandemic, how do rates of depression among SHCs compare with national individual averages acquired in the U.S. Census Bureau’s Household Pulse Survey (HPS) data?

## 2. Materials and Methods

### 2.1. Study Overview

At the start of the COVID-19 pandemic, the Emory Department of Spiritual Health deployed a daily emotional tracking tool intended to foster a momentary pause to reflect on one’s own well-being and to direct staff to resources for help if needed. The emotion tracking tool was presented to staff as a survey to monitor their own levels of burnout, depressive symptoms, and well-being. A link to the daily emotion pulse survey was emailed to SHCs, researchers, and staff associated with the spiritual health department every weekday morning at 7:00 am (measures described below) from 30 March 2020 until 30 April 2021. The early morning reminder and link to the survey were intended as an opportunity for each person to check in with themselves at the start of their workday and consider whether they were approaching overwhelm beyond their capacity for resilience. Individuals could elect to complete the survey daily, occasionally, or not at all. As the survey was intended as an anonymous self-monitoring tool, data were not collected on demographics, work conditions, or background experience. The survey used validated and commonly used measures and was intentionally brief to allow for ease of use and rapid completion at the start of the workday. Each person on the survey distribution list also received an auto-generated emailed report every Saturday that graphically depicted their trend in well-being, showing their running weekly average scores for however many surveys were completed in each week. Each weekly report provided information on how to access mental healthcare resources. 

Although the daily surveys were not designed to collect data on potential contributors or impediments to well-being, they were recognized as valuable epidemiological data for understanding the emotional experience of SHCs and how it might be changing over the course of the COVID-19 pandemic. Therefore, the study team obtained approval from the Emory University Institutional Review Board to use the pulse data and retroactively obtained consent from the SHCs for use of the data. This study included data from the tracking tool rollout on 30 March 2020 through 20 December 2020, as there were gaps in data completion over the Christmas and New Year holidays and few SHCs used the tool in the new year. No incentives were provided to use the tracking tool or to be included in this study. We did not include pulse surveys acquired from researchers or staff who were not involved in frontline spiritual care in this study. Only data from SHCs who provided consent were used. There were no exclusion criteria.

### 2.2. Sample

SHCs in this study play a crucial role within five inpatient hospital locations in a major urban healthcare system in the southeast U.S., providing support and consultation to patients from various backgrounds, regardless of faith tradition (or lack thereof). In general, SHCs in this system aim to consult with all inpatients upon their admission to the hospital. Additionally, consultations are conducted based on patient or family requests, and requests from clinical staff if they have concerns about a patient’s well-being. SHCs also respond to cardiac arrest codes and deaths, often providing spiritual support to patients, families, and staff during these difficult situations. SHCs facilitate discussions around advance directives and end-of-life planning, helping patients and their families navigate complex decisions regarding treatment preferences and the final stages of life. During 2020, SHCs in this healthcare system conducted 75,751 consults with patients and their loved ones, and 87,006 consults with healthcare staff.

During the early weeks of the pandemic, SHCs were not able to go into COVID-19 rooms (primarily because of a lack of personal protective equipment [PPE]). However, SHCs continued to provide care to COVID-19 patients and to their at-home family by telephone. Similarly, SHCs were initially not able to provide face-to-face care for staff working in COVID-19 units but provided consults to staff by telephone. SHCs continued to provide uninterrupted care for patients and staff in the rest of the hospital. When PPE became more available and after the COVID-19 vaccine rollout, SHCs were able to see COVID-19 patients in person. However, the timeframe under focus in this study occurred prior to vaccine availability. Forty-five SHCs received the daily emotional tracking tool and were retrospectively contacted to consent to participating in this study. Of these, 32 (71%) consented and participated at least once in the survey. However, SHCs completed surveys erratically. The most frequent users (*n* = 4) completed surveys in 27 or 28 weeks of the 38-week study period, averaging 3 to 4 days per week. The next most frequent group (*n* = 11) completed surveys in 8 to 16 of the weeks, averaging 1 to 4 days per week. Eleven chaplains completed fewer than 10 surveys each, using the tool in only 1 to 4 weeks.

### 2.3. Measures

At rollout, the daily emotional tracking tool enabled SHCs to self-report on burnout, depression, and well-being. We measured burnout using the single-item burnout from the Mini-Z burnout survey [16], which has been validated as a measure of the emotional exhaustion dimension of burnout in healthcare settings and compares favorably to other longer measures of burnout [17,18,19]. Participants are queried to use their own definition of burnout and to rate their burnout from 1 = “I enjoy my work. I have no symptoms of burnout” to 5 = “I feel completely burned out and often wonder if I can go on. I am at the point where I may need some changes or may need to seek some sort of help.”

Depression was measured using Patient Health Questionnaire PHQ-2, a validated two-item measure used to screen for symptoms of major depression [20]. The measure includes questions about the frequency of depressed mood and anhedonia, each of which is rated on a 4-point scale from 0 = not at all to 3 = nearly every day. Although the PHQ-2 was designed to query depression symptoms over the previous two weeks, here we asked respondents to reflect on their feelings over the previous day. Total score ranges from 0 to 6, with positive screening for depression commonly being identified at a cutoff point of 3. The scale demonstrated good reliability in our sample (Cronbach’s alpha = 0.84).

We used the World Health Organization Well-Being Index (WHO-5) to measure self-rated psychological well-being [21]. Although the WHO-5 was designed to query well-being over the previous two weeks, here we asked respondents to reflect on their feelings over the previous day. The WHO-5 includes 5 items (e.g., “I have felt cheerful and in good spirits”), with each item rated on a scale of 0 = at no time to 5 = all of the time. Rating totals are multiplied by 4 to yield a scale score ranging from 0 (least well-being) to 100 (most well-being). The scale demonstrated good reliability in our sample (Cronbach’s alpha = 0.89).

Daily COVID-19 rates were compiled from the electronic health record across Emory’s hospital system for admissions of patients with COVID-19, hospital deaths of COVID-19 patients, and the number of patients with COVID-19 in the intensive care unit (ICU). To compare depression rates in our sample with national averages, we used data from Phase 1 to Phase 3 (waves 1–21) of the Household Pulse Survey (HPS), a survey of U.S. households drawn from the U.S. Census Bureau’s master file of approximately 117 million housing units [22]. The HPS, offered to a nationally representative sample of over 50,000 adults, collected data on a consistent weekly or biweekly basis with breaks during holiday seasons. The national survey pulses began in April 2020, nearly coincidental to our data collection start date of March 2020, and continued through the end of our study period. Although the HPS did not collect data on burnout or the WHO-5 measure of well-being, it did collect data on depression using the same measure we chose, the PHQ-2, making it feasible to directly compare our data on depression with national levels. 

### 2.4. Analysis

To characterize burnout, depression, and well-being among SHCs in the first year of the COVID-19 pandemic, we employed descriptive statistics to present biweekly averages for each outcome. Multi-level logistic regression was used to examine the relationship between COVID-19 rates (admissions, deaths, ICU census) and each outcome (9 total associations), while accounting for clustering in the data by SHC. Because outcomes (and residuals) were not normally distributed, as determined by the Shapiro–Wilk test, and because attempts to achieve normality through common data transformation techniques for positively skewed data (Log(10), inverse, and square root) failed, our data did not meet the assumptions required for linear regression. To run logistic regression, each outcome was dichotomized at “none” vs. “any” negative impact to mental health; that is, burnout score of 1 vs. 2–5; PHQ-2 score of 0 vs. 1–6; WHO-5 prorated score of 80–100 vs. <80. These cutoff points were selected largely due to the severe skew in the data, so that the group sizes would be as close to equal as possible to minimize classification bias from unbalanced data. Both intercept (logit of the baseline outcome score) and slope (logit of the degree to which changes in the COVID-19 rate predict changes in the outcome) were allowed to vary as random effects in each model. 

Spearman correlation tests were conducted to assess correlation between each outcome and each COVID-19 metric (9 total associations, based on daily averages) to confirm regression results. Correlations were further tested using a one-week delay in outcome data to explore the possibility of a delayed effect of COVID-19 rates on SHC burnout, depression, and well-being. The analyses described above were completed using all SHC-provided data, regardless of how many surveys were completed by each individual. Correlation tests were repeated while limiting the data to those from SHCs with at least 10 data points over time to mitigate the potential for bias in the data. A third of the regressions were spot-checked using this reduced dataset. 

Finally, using descriptive statistics, biweekly mean PHQ-2 scores were compared with national means. We further compared the change in mean PHQ-2 score with the change in national mean score across phases of the HPS. 

## 3. Results

A total of *n* = 840 tracking tool surveys were completed by 32 SHCs, who each completed between 1 and 120 surveys (mean = 37.3, standard deviation [SD] = 16.1). Mean average burnout score across the nine months of the study was 1.73 (SD = 0.70); mean PHQ-2 score was 0.41 (SD = 0.95); and mean WHO-5 score was 76.2 (SD = 17.0). Biweekly means are presented in Table 1 and Figure 1. Of the 840 entries, 90.0% indicated no symptoms of burnout, with 8.3% of entries indicating one or more symptoms of burnout, 1.3% indicating symptoms of burnout that would not go away, and 0.4% indicating complete burnout. No biweekly mean score met the cutoff for burnout. Of the 840 entries, 97.1% were below the cutoff for depression (<3), and no biweekly mean score met the cutoff for depression.

We ran nine logistic regression models to evaluate the relationship of each outcome (burnout, depression, and well-being) to each COVID-19 metric (hospital admissions, hospital deaths, ICU census). Regression models were run independently to maximize precision and reduce effects from multicollinearity, as input (and COVID-19 metric) variables were significantly, and often strongly, correlated. Results from each regression illuminated three important points. First, was the degree to which variability in outcome measures was based on differences across SHCs. To determine this for each outcome, running a null model containing the intercept only and allowing it to vary across clinician produced an intraclass correlation (ICC) that represented the proportion of variance in the outcome that can be attributed to differences between clinicians. Based on the ICCs, 52.8% of the variation in burnout was attributable to the effect of clinician, attribution for depression was 30.1%, and for well-being it was 46.3%. Second, outcome levels did not rise or fall with COVID-19 rates. The results from each of the nine multi-level logistic models are presented in Table 2. There was no statistically significant relationship between any of the predictors (COVID-19 admissions, hospital deaths from COVID-19, or COVID-19 ICU census) and any of the outcomes (burnout, depression, or well-being), as shown in the fixed effects section of Table 2. Third, the full regression models confirmed statistical significance of the differences in outcome levels across SHCs but showed no difference in the effect of COVID-19 metrics on those levels across SHCs, as shown in the random effects section of Table 2. For example, the regression evaluating the link between COVID-19 admission rates and burnout showed that burnout level varied significantly across SHC, but the effect of COVID-19 admissions on burnout did not vary across SHC.

Spearman correlation tests were conducted to assess correlation between each outcome and each COVID-19 metric (nine total associations, based on daily averages) to confirm regression results. This allowed for a check on the regression findings while being able to use the full range of variability in the data, compared with the dichotomized outcomes used in the regressions. The evaluation of Spearman correlations between each predictor and each COVID-19 outcome did not identify any significant associations in the data as collected. Further, when data were analyzed with a one-week delay in outcome, only COVID-19 admission rate was associated with burnout one week later (r_s_ = −0.187, *p* = 0.008); however, this analysis did not take into account differences between clinicians who may have contributed different quantities of data to the dataset or contributed data at different points of the first-year trajectory. No substantial difference in correlation or regression results were identified when excluding data from SHCs who completed fewer than 10 surveys. Because it accounts for variation across SHCs, the regression analysis was considered the most revealing of predictor effects on outcomes.

Nationally, mean depression levels as measured by the PHQ-2 rose 20.6% (from 1.59 to 1.92) from the beginning to the end of Phase 1 of the Household Pulse Survey (Table 3). During that same period, mean depression levels in our SHC sample dropped from 0.77 to 0.38, a 49.8% decrease. From the beginning of Phase 2 through the end of our study period, national mean depression levels rose 19.1% (from 1.64 to 1.95), while our SHC sample showed a 1.2% decrease in mean depression levels, with PHQ-2 scores just below 0.24 at both comparison points. Mean national PHQ-2 scores during each weekly or biweekly survey wave across Phase 1 and Phase 2 were consistently higher than our sample’s biweekly means (Figure 2).

## 4. Discussion

The COVID-19 pandemic altered lifestyle and social behaviors and led to disruptive changes to all aspects of society, with the potential for harmful impact on mental health symptoms. Additional challenges for those working in healthcare led to even higher rates of mental health problems for those on the frontlines of care [2]. In this study, which used daily pulse surveys to assess burnout, depression, and well-being among SHCs working in a major urban hospital system during the first year of the pandemic, we found that their burnout, depression, and well-being were unrelated to local COVID-19 rates. Moreover, SHCs reported lower depression rates than the general population, and appeared to be buffered from an increase in depression reported by a nationally representative sample of adults surveyed during the same period. 

These data contribute to important insights about the mental and professional health and well-being of frontline healthcare workers during a crisis. Overall, the studies that have emerged indicate that burnout rates during the pandemic were highly heterogenous across contexts, roles, and specialties. One large recent meta-analysis of 45 observational studies totaling almost 30,000 healthcare professionals around the world found that the pooled prevalence of burnout among healthcare workers was 54.6%, with rates roughly equal between physicians and nurses [23]. However, another recent systematic review and meta-analysis of 34 cross-sectional studies estimated that the prevalence of burnout among physicians during the COVID-19 pandemic varied widely, ranging from 6.0% to 99.8% [1]. The highly discordant findings and variable rates are due in part to heterogenous contextual factors that confer protection from, or risk of, burnout. For example, cross-sectional data acquired from over 26,000 public health workers during March and April 2021 indicate that isolation was associated with an increased risk of depression, whereas the ability to take time off was protective [24]. Additionally, frontline workers appear to have had higher rates of burnout than second-line workers (defined as healthcare workers not directly working with COVID-19 patients) [23]. 

While we do not have data for our SHC sample prior to the pandemic, rates of burnout and depression were exceedingly low in our study. Although previous studies have shown low levels of burnout and a high degree of resiliency among SHCs, the low levels of burnout in the early stages of COVID-19 were not expected. There were reasons to hypothesize that the pandemic may have been particularly harmful to SHCs, in part due to protocol changes that may have removed or reduced many of the interpersonal factors that were protective. Research conducted with SHCs at other sites during the pandemic indicate that they experienced increased disconnection, depersonalization, and uncertainty, all of which would be expected to lead to increased burnout and mental health symptoms [14,15]. We did not systematically query SHCs in our study about these factors, and it is possible that the low rates of burnout and depression reflect relatively more social support, clarity of roles, or personal meaning than were experienced by SHCs in other healthcare systems. Alternatively, SHCs in our study may have maintained their professional and personal well-being even in the face of increased risk factors. 

It is important to examine our findings on SHC well-being not only within the context of healthcare, but also considering what is known about the impact of the pandemic on population-level well-being. With a large number of reports, pooled prevalence estimates of depression during the pandemic varied widely [25,26]. Two large meta-analyses (at least 50,000 individuals) reporting on pooled effect sizes reached the consistent conclusion that the pandemic’s effects on depression were small and generally most pronounced early in the pandemic, with rates of mental health symptoms returning to (or close to) pre-pandemic levels by the middle of 2020 [27,28]. Similarly, suicide trends in 21 middle- and high-income countries indicated that there was not an increase in suicides during the early stages of the pandemic; in fact, rates fell in 12 countries, including in the USA [29]. Importantly, however, the effects appear to have been highly varied, with women and young adults consistently reporting larger increases in mental health symptoms [26]. Moreover, the pandemic increased socio-cultural and economic disparities, with the likelihood of greater harmful effects on mental health among groups experiencing increased hardship and structural violence [30,31,32,33].

Our findings are consistent with studies showing that rates of burnout were highest early in the pandemic. A recent meta-analysis found that burnout rates were highest during the first months of the pandemic (60.7%, from March to August 2020) compared with later in the pandemic period (49.3%, September 2020 to June 2021) [23]. A meta-synthesis of qualitative studies conducted during the first year of the COVID-19 pandemic and during other pandemics (e.g., MERS, Ebola) found that safety concerns were highest during the early phases of COVID-19 and other pandemics, especially when protective equipment, information, and resources were inadequate [34]. Related, this synthesis identified healthcare providers facing the ethical and moral dilemma of protecting their own safety at the expense of providing their usual standard of quality patient care as a primary source of tension. SHCs in our study certainly experienced this tension during the initial weeks of the pandemic, with a chaotic environment and various changes and limitations to their usual patient care. SHCs were initially not able to enter rooms or units with COVID-19 patients, and switched much of their care to tele-chaplaincy, support of family no longer able to visit patients, and support of staff stressed by fears around COVID-19 and an excessive number of patient deaths, often in isolation.

Although rates of depression and burnout were highest at the start of the pandemic, SHCs in our study did not report increases of the magnitude observed in the general public during the early months. These data were collected during a time when national trends in anxiety and depression were in flux. From August to December 2020, the average anxiety and depression scores for adults increased by 13% and 14.8%, respectively, as assessed by U.S. Census Bureau survey data. From December to June 2021, anxiety and depression scores among the general public dropped. However, rates of burnout and depression began dropping immediately after commencement of pulse data collection among the SHCs in our study. Moreover, SHC burnout and well-being were not related to any of the COVID-19 acuity metrics in their hospital system, further highlighting the apparent resilience of SHCs even in the face of crisis. In general, healthcare chaplains derive a high level of meaning from their work, and it is likely that the pandemic also resulted in personal and professional growth. The experience of increased personal growth has been reported during other pandemics, as consistently documented in qualitative research [34]. Notably, the hospital system that was the context of this investigation continued a high level of commitment from the spiritual health clinicians, even while many other hospital systems prohibited their healthcare chaplains from working on COVID-19 units or even within the hospital [8]. A phrase often used throughout the pandemic was that Emory spiritual health was committed to “being in the room”. Policies resulting from this commitment made clear the essential role of the discipline of spiritual health within the healthcare organization and arguably bolstered the mental and workplace well-being of the SHCs during the pandemic, especially when viewed alongside evidence that many healthcare chaplains elsewhere experienced restrictions and feelings of confusion about (and devaluation of) their role during the pandemic [8].

Our finding that SHC burnout and well-being was not statistically related to local COVID-19 rates also contributes to the discussion of the nature of compassion fatigue, a term often used to describe an acute reduction in compassionate feelings towards others in response to occupational stress among healthcare providers [35,36]. The concept of compassion fatigue has been highly criticized in several comprehensive reviews, in part due to the consistent observation that increased opportunities to experience and express compassion seem to beget more (not less) compassion, and bolster the well-being of the provider, especially in the face of elevated workplace stress [35,36]. Previous research conducted with healthcare chaplains supports this critique of the compassion fatigue construct, with ample evidence indicating that chaplains, whose essential role in the organization is to deliver compassion, experience lower rates of burnout and higher compassion satisfaction than other healthcare provider types, despite working with high-acuity patient populations and often in times of trauma, loss, and crisis [11,12,13]. Although we did not directly measure compassion or compassion fatigue, our findings are consistent with the idea that SHC well-being is not impaired by providing care to severely ill patients, even in a high-stress environment during an extended crisis. 

Limitations. A potential limitation of our study is that SHC responses may have been influenced by self-report biases such as a social desirability bias, especially with an implicit and explicit expectation of some legacy norms of chaplaincy in pressuring to care for others without limit and without personal struggle. The potential for bias is somewhat mitigated based on the establishment of the pulse survey as a tracking tool for SHCs to privately and anonymously monitor their own well-being; it was not originally slated to be part of a study and there was no expectation that these data would be viewed by others or used in any way beyond personal information for the SHC. A second limitation is that data completion may not be random; thus, our findings may be influenced by pulse survey completion patterns. For example, SHCs who felt better or were less fatigued may have more consistently completed the daily pulse surveys. The fact that our findings do not substantively change, even when low-response participants are excluded, mitigates this concern. Moreover, although it is likely that important second-level predictors such as SHC age, gender, and years in the role may predict daily well-being indicators, we are unable to conduct those analyses given the nature of the dataset. A third limitation is that we used a brief single-item burnout measure. Although this item has been validated as a measure of the emotional exhaustion dimension of burnout [16,17,18,19], we may have missed important aspects of the experience of burnout by not measuring other dimensions of burnout such as depersonalization or personal accomplishment. Related, it is possible that these other burnout dimensions may have more closely tracked local COVID-19 acuity. Although we would hypothesize that the emotional exhaustion dimension would be most correlated with clinical demands, we cannot test that in this study. Finally, these findings are from a limited number of SHCs providing spiritual consultations in one hospital system and, importantly, one that provides compassion training specifically to improve resilience among SHCs [37]. These data may not be representative of healthcare chaplains working in other systems. Moreover, the small sample size may have biased the multi-level modeling estimates, though concerns that this influenced our findings are mitigated by the fact that we had a similar pattern of findings in our correlation analyses. Despite these limitations, the current study may represent a starting point for future research to identify the factors that contribute to preventing burnout and depression in SHCs and other healthcare providers.

## 5. Conclusions

Ultimately, these unique data collected daily during the first year of the pandemic yield findings that may inform what is known about preventing burnout and improving flourishing among healthcare providers. Perhaps the most important implication of our findings is that SHC mental health and professional well-being were buffered from an acutely harmful impact of pandemic care during the crisis and appear to have been bolstered in as much as the most concerning scores occurred in the very earliest days of the pandemic. When it comes to witnessing the tragedy of a healthcare crisis, our findings indicate that those who are able to provide compassionate responses may be most protected against burnout and depression.

## Figures and Tables

**Figure 1 ijerph-21-00944-f001:**
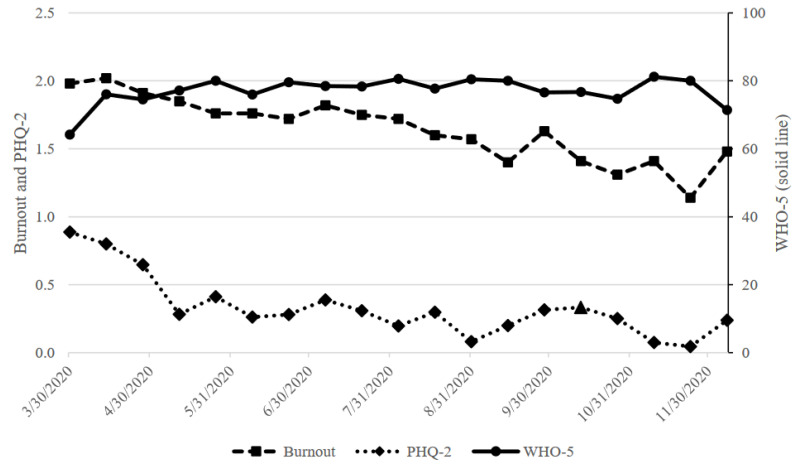
Mean biweekly outcomes (WHO-5 on right axis).

**Figure 2 ijerph-21-00944-f002:**
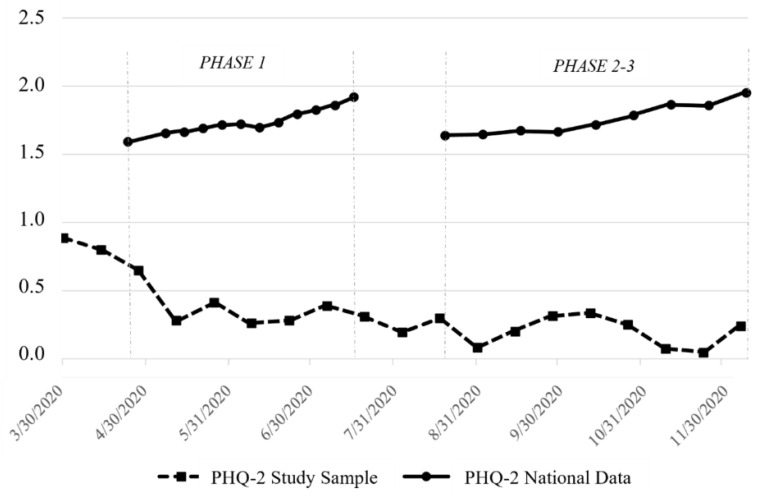
Comparison of national and study sample mean PHQ-2 scores.

**Table 1 ijerph-21-00944-t001:** Biweekly mean burnout, depression, and well-being scores.

Period Start Date	Mean Burnout Score			Mean PHQ-2 Score			Mean WHO-5 Score		
	Mean	SD	Median	Mean	SD	Median	Mean	SD	Median
30 March 2020	1.98	0.87	2.0	0.89	1.61	0.0	64.21	27.54	76.0
13 April 2020	2.02	0.70	2.0	0.80	1.42	0.0	76.06	18.67	76.0
27 April 2020	1.91	0.81	2.0	0.65	1.18	0.0	74.53	18.06	76.0
11 May 2020	1.85	0.67	2.0	0.28	0.65	0.0	77.13	15.65	80.0
25 May 2020	1.76	0.44	2.0	0.41	1.46	0.0	80.00	13.11	80.0
8 June 2020	1.76	0.71	2.0	0.26	0.53	0.0	76.00	11.28	76.0
22 June 2020	1.72	0.73	2.0	0.28	0.67	0.0	79.57	11.48	80.0
6 July 2020	1.82	0.72	2.0	0.39	0.65	0.0	78.45	9.92	80.0
20 July 2020	1.75	0.50	2.0	0.31	0.55	0.0	78.35	10.37	80.0
3 August 2020	1.72	0.54	2.0	0.20	0.54	0.0	80.61	9.31	80.0
17 August 2020	1.6	0.58	2.0	0.30	0.66	0.0	77.70	10.44	80.0
31 August 2020	1.57	0.56	2.0	0.08	0.36	0.0	80.43	13.12	80.0
14 September 2020	1.4	0.50	1.0	0.20	0.62	0.0	80.00	21.29	80.0
28 September 2020	1.63	0.60	2.0	0.31	0.63	0.0	76.57	16.81	80.0
12 October 2020	1.41	0.69	1.0	0.33	0.68	0.0	76.74	19.97	80.0
26 October 2020	1.31	0.48	1.0	0.25	0.68	0.0	74.75	19.42	80.0
9 November 2020	1.41	0.50	1.0	0.07	0.38	0.0	81.19	14.24	80.0
23 November 2020	1.14	0.35	1.0	0.05	0.21	0.0	80.00	14.45	80.0
7 December 2020	1.48	0.68	1.0	0.24	0.54	0.0	71.43	17.59	80.0

**Table 2 ijerph-21-00944-t002:** Fixed and random effects in multi-level logistic regression of burnout, depression, and well-being outcomes on hospital COVID-19 rates.

	Fixed Effects							Random Effects					
Outcome—Predictor	Coeff.	Std. Error	t	Sig.	Exp	95% Conf Interval		Estimate	Std. Error	Z	Sig.	95% Conf. Interval	
Burnout—admissions	−0.001	0.0098	−0.075	0.940	0.999	0.980	1.019	0.000 ^a^					
Burnout—deaths	0.040	0.0852	0.471	0.638	1.041	0.881	1.230	0.011	0.031	0.344	0.731	3.56 × 10^−5^	3.201
Burnout—ICU census	0.007	0.0381	0.178	0.859	1.007	0.934	1.085	0.009	0.011	0.791	0.429	0.001	0.102
PHQ-2—admissions	0.001	0.0141	0.049	0.961	1.001	0.973	1.029	0.001	0.001	0.861	0.389	0.000	0.011
PHQ-2—deaths	−0.019	0.0718	−0.268	0.789	0.981	0.852	1.129	0.002	0.018	0.122	0.903	2.32 × 10^−10^	19,987
PHQ-2—ICU census	0.034	0.0351	0.973	0.331	1.035	0.966	1.109	0.007	0.007	0.946	0.344	0.001	0.055
WHO-5—admissions	0.016	0.0121	1.338	0.181	1.016	0.992	1.041	0.001	0.001	1.311	0.190	0.000	0.003
WHO-5—deaths	0.080	0.0685	1.167	0.244	1.083	0.947	1.239	0.000 ^a^					
WHO-5—ICU census	0.004	0.0282	0.136	0.892	1.004	0.950	1.061	0.002	0.003	0.810	0.418	0.000	0.027

^a^: not enough variability for model to successfully run.

**Table 3 ijerph-21-00944-t003:** Comparison of national with sample PHQ-2 score changes across Phase 1 and 2 of the U.S. census household pulse survey.

Outcome Measure	Household Pulse Survey Phase 1			Household Pulse Survey Phase 2		
	23 April–5 May 2020	16–21 July 2020	% Change	19–31 August 2020	9–21 December 2020	% Change
National mean PHQ-2 score	1.59	1.92	20.6%	1.64	1.95	19.1%
SHC mean PHQ-2 score	0.77	0.38	−49.8%	0.24	0.24	−1.2%

## Data Availability

De-identified data that support the findings of this study are available on request from the corresponding author, J.S.M.

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
