# Peer review of "Hospital Chaplain Burnout, Depression, and Well-Being during the COVID-19 Pandemic"

_ijerph, 2024, doi:10.3390/ijerph21070944_

Round 1

Reviewer 1 Report

Comments and Suggestions for Authors

I really enjoyed reading this manuscript and the results were impressive.

There are several things I would like to mention that may help improve this manuscript.

1.       The first research question is not actually research question. I don’t quite understand the use of ‘characterize’ in the first question. In addition, I don’t see any results and insights related to this question.

2.       The authors mentioned that each outcome was dichotomized at the mean. But the distribution of the mean is not normal so mean scores for each outcome may not be feasible. Please provide more information why the authors chose the mean scores. Another thing is that the single-item burnout question has five response options, and the first two response options could be categorized as ‘not burned out’, but the remaining three could be ‘burned out’. The authors should provide more information on why they chose the mean score to dichotomize each outcome.

3.       The authors didn’t provide any demographic characteristics of the sample.

4.       It was found that variations were largely from clinicians (second-level). In this case, I believe that the authors should look at predictors of the second-level, such as demographic characteristics, years of work, and so on.

5.       The authors need provide more information on why they ran nine logistic regression analyses separately instead of running multivariate logistic regressions that included three predictors (burnout, depression, well-being) together. Including predictors together could benefit from examining interactions among predictors, which might be more interesting.

6.       I think the Spearman correlation results were related to the first research question. If so, please move this result to the first section of the results to be consistent with the order of research questions.

7.       Related to this, the authors used continuous variables here, instead of the categorical ones used in the logistic regression for the second research question. This could be confusing. If the authors decided to use categorical variables, then these variables should be used consistently across all research questions.

8.       If the authors found that the results of the current study were different from previous findings, they should discuss why or how the SHCs in the study were different. The authors did mention this in the paragraphs on page 8, lines 245-252 but it needs to elaborate further so that the audience can gain insight from this.

9.       When comparing to national data, the authors need to provide more information to understand whether the national sample has the same or similar conditions to the current study sample. Could the authors consider other variables that affect the outcomes before making a comparison?

Reviewer 2 Report

Comments and Suggestions for Authors

Comments and suggestions for authors

I read the manuscript entitled ‘Fluctuation in hospital chaplain burnout, depression, and well-being during the COVID-19 pandemic’. The impact of COVID pandemic on hospital chaplain represent an interesting topic which was less studied compared to that on doctors and nurses.

Introduction is clearly presented, but it could be expanded.

Material and method

Authors should improve this section and explain clearly the procedure.

Lines 64-72 - about ‘daily emotional tracking tool’. Why was this tool sent ‘every day, at 7 am’? Wasn’t it the same? When the spiritual health clinicians (SHCs) sent their responses? They sent their responses daily?  As I understand, they received weekly a report about their trend in wellbeing, which assume a number of responses per week from each participant.

Sample: lines 81-102. There is a description of the activities of SHCs. In this section it should be specified the number of participants included in the study. Which is the number of SHCs who received the daily emotional tool? Where there any inclusion/ exclusion criteria? Which was the response rate? How many of them (or percentage) completed the emotional tool and sent it daily/ at least weekly? Some of these data are mentioned in the Result section, line 159 (‘32 SCHs completed a total of 840 tracking tool surveys’). At line 160: ‘Each completed between 1 and 120 surveys’. The participant(s) which completed 1 survey was taken into consideration? At lines 209-210 is mentioned that there were ‘excluded data from SHCs who completed fewer than 10 surveys’. How many participants were in this situation? Also, from line 160, I deduced that at most a survey was sent every 2 days and only by a few of SHCs.  A description of the sample would also have been useful (sociodemographic data: age, gender, seniority in the profession, etc.) and perhaps it would have been suitable to be considered in the statistical analysis.

Measures. Authors should specify why they choose to apply the mentioned questionnaires with 1 respective 2 questions; maybe for ease of application. But did the authors choose PHQ-2 due the fact that the same questionnaire was used in Household Pulse Survey? As is written in Result section (lines 191-199) the authors made a comparison with the level of depression in the general population during the pandemic.

Results section

Lines 174-178 – Authors should explain more clearly this paragraph. Also, the word ‘clinicians’ is confusing. Reading further, it seems to be about SHCs.

Although the statistical analysis appears to be solid, the results are based on a subjective evaluation of the level of burnout, depression and well-being of a small or limited number of subjects. 

Discussion section

The research should have started with hypotheses, establishing input and output variables. The authors mention that they collected daily surveys which were recognized as valuable data(line 73) and then they thought of doing a study.

The study would have been more interesting if it had researched or identified which factors or variables may be related to lower levels of burnout and depression and to a higher state of well-being at SHCs. The discussions seem largely speculative (e.g. lines 249-252) and are not supported by the results.

Also, there seem to be some contradictions (e.g. lines 241-245  vs lines 41-43   and lines 317-320)

The study’s limitations are correctly presented, but they are not the only ones. The study seems to have no direction, no purpose, it only identifies the low values of burnout and depression in SHCs, without any finality or practical implication.

Conclusions section

Unfortunately, the study does not ‘inform what is known about preventing burnout and improving flourishing among healthcare providers’. The findings of the study don’t indicate the role of ability to provide compassion regarding the levels of burnout and depression (in lines 320-321 authors wrote ‘Although we did not directly measure compassion or compassion fatigue’).

The current study may represent a starting point for future research to identify the factors that contribute to prevent burnout and depression in SHCs

After reading the manuscript and considering the research results, maybe the term ‘fluctuation’ in the title is not the most appropriate.

References: Out of a total of 40 references, 15 are prior to 2018.

Date: 10.05.2024

Reviewer 3 Report

Comments and Suggestions for Authors

Thanks for the opportunity to read and review this manuscript. This manuscript presents a cross-sectional study examining self-report burnout, depression and well-being in a sample of spiritual health clinicians (SHCs) using data from a daily pulse survey during the first year of the COVID-19. Overall the research topic is timely and important. The method is sound and the sample sufficiently large to uncover statistical relationships. The manuscript was well written with necessary details provided. The study findings add to the emerging evidence characterizing burnout and mental health of frontline healthcare professionals during different phases of the pandemic. I believe these findings will be of interest to the readers of the journal and therefore I recommend the manuscript for publication.There is only a minor point which I would encourage the authors to take into consideration in the revision:

Due to the nature of the pulse survey, psychological constructs were measured using very brief scales. Particularly, burnout was assessed using only one item. The usage of the one-item measure should be better justified and the potential bias carefully discussed. In addition, the present study found a very low burnout rate compared to other studies examining burnout of healthcare workers during the pandemic. However, most of the previous studies have used scales with multiple items and dimensions to assess burnout (e.g., Alkhamees et al., 2023; Macaron et al., 2023 ). I would encourage the authors to make further discussions on how the differences in measures might contribute to the inconsistencies in the findings.

Reviewer 4 Report

Comments and Suggestions for Authors This paper examines self-assessed levels of burnout, depression, and well-being among spiritual health clinicians working within five inpatient hospitals during the Covid-19 pandemic.     The paper is coherently structured, well written, and makes a clear argument in relation to the evidence. While a number of the reported results are not significant, the authors provide evidence to demonstrate that these results are largely against expectations. As a result, I believe the paper makes an original contribution to the literature.   I would only make two comments in relation to the paper.   Although they do feature elsewhere, the actual sample sizes do not appear to be reported in section 'ii. Sampling'.    In the limitations section, I wonder whether there should be some discussion on the impact of sample sizes in multilevel modelling. The mention of the case-study design is also very brief and perhaps a more robust defence could be made: is there any reason to think that this particular hospital system is different to others? 
